# Dual-Layer Spectral Detector Computed Tomography Quantitative Parameters: A Potential Tool for Lymph Node Activity Determination in Lymphoma Patients

**DOI:** 10.3390/diagnostics14020149

**Published:** 2024-01-09

**Authors:** Hebing Chen, Yuxiang Fang, Jin Gu, Peng Sun, Lian Yang, Feng Pan, Hongying Wu, Tianhe Ye

**Affiliations:** 1Department of Radiology, Union Hospital, Tongji Medical College, Huazhong University of Science and Technology, Jiefang Avenue #1277, Wuhan 430022, China; hebing1027898473@126.com (H.C.); yuxiangfangxh@163.com (Y.F.); gujin-ll@163.com (J.G.); yanglian@hust.edu.cn (L.Y.); uh_fengpan@hust.edu.cn (F.P.); 2Hubei Province Key Laboratory of Molecular Imaging, Jiefang Avenue #1277, Wuhan 430022, China; 3Clinical & Technical Support, Philips Healthcare, Floor 7, Building 2, World Profit Center, Beijing 100000, China; peng.sun@philips.com

**Keywords:** dual-layer, spectral detector, effective atomic number, iodine concentration, electron density, extracellular volume, lymphoma, lymph nodes

## Abstract

Dual-energy CT has shown promising results in determining tumor characteristics and treatment effectiveness through spectral data by assessing normalized iodine concentration (nIC), normalized effective atomic number (nZeff), normalized electron density (nED), and extracellular volume (ECV). This study explores the value of quantitative parameters in contrast-enhanced dual-layer spectral detector CT (SDCT) as a potential tool for detecting lymph node activity in lymphoma patients. A retrospective analysis of 55 lymphoma patients with 289 lymph nodes, assessed through ^18^FDG-PET/CT and the Deauville five-point scale, revealed significantly higher values of nIC, nZeff, nED, and ECV in active lymph nodes compared to inactive ones (*p* < 0.001). Generalized linear mixed models showed statistically significant fixed-effect parameters for nIC, nZeff, and ECV (*p* < 0.05). The area under the receiver operating characteristic curve (AUROC) values of nIC, nZeff, and ECV reached 0.822, 0.845, and 0.811 for diagnosing lymph node activity. In conclusion, the use of g nIC, nZeff, and ECV as alternative imaging biomarkers to PET/CT for identifying lymph node activity in lymphoma holds potential as a reliable diagnostic tool that can guide treatment decisions.

## 1. Introduction

Rapid advancements in highly sensitive and specialized technologies for disease assessment and therapeutic approaches, including chemotherapy, radiation therapy, targeted therapy, immunotherapy, CAR T-cell therapy, and stem cell (bone marrow) transplantation, have significantly improved the survival rates of lymphoma patients in recent decades. Quantitative and functional imaging techniques, including computed tomography (CT) and 2-deoxy-2-[^18^F]fluoro-D-glucose (FDG)-positron emission tomography (PET)/X-ray computed tomography (CT), now play a crucial role in staging before treatment, interim evaluation, and post-treatment assessment for lymphoma, surpassing established clinical risk factors [1,2,3].

For lymph node evaluation, conventional CT provides accurate measurement of nodal size, e.g., longest diameter and shortest diameter, which is an important characteristic for differentiating benign and malignant lymph nodes [4,5,6] since larger lymph nodes were more likely to be malignant. However, the diagnostic accuracy of conventional CT is limited for lymphoma due to a lack of functional or metabolic information [3,7].

Across different histologies, 93% to 99% of lymphomas exhibit ^18^FDG avidity, with Hodgkin’s lymphoma, diffuse large B-cell lymphoma (DLBCL), follicular lymphoma, Burkitt lymphoma, and mantle cell lymphoma being the most FDG avid tumors [3]. Compared to CT, ^18^FDG-PET/CT can assess lymph nodes’ metabolic and proliferative activity by quantitatively measuring metabolic tumor volume and total lesion glycolysis. This enhances the accuracy of staging and treatment response evaluation [8]. Previous studies have demonstrated that in HL and ^18^FDG-avid non-Hodgkin’s lymphoma subtypes, PET and PET/CT scans enhance staging accuracy compared to CT scans, especially for nodal and extranodal sites [8]. PET/CT leads to a stage change in 10% to 30% of patients, with upstaging occurring more frequently, although changes in management are less common [3]. PET/CT scanning has become the standard imaging technique for assessing treatment response in most lymphoma cases [9]. However, some researchers have also suggested that the metabolization of tumors in response to treatment is slow, leading to potentially false positive results within weeks or months, making ^18^FDG-PET/CT controversial for interim response assessment in lymphoma [3,10]. Additionally, it should be noted that ^18^FDG-PET/CT also has other limitations, such as high examination expenses, substantial radiation exposure, and restricted patient access [11].

In recent decades, CT technology has evolved from single-energy CT to dual-energy CT [12,13]. Dual-energy CT, or spectral CT, generates material-specific images based on atomic number and unique mass attenuation coefficients at different X-ray energies, improving lesion detection and characterization [12,14]. The latest innovation, dual-layer spectral detector CT (SDCT), utilizes a single X-ray tube and two detector layers to capture low- and high-energy photons simultaneously [13]. SDCT provides spectral data for a range of conventional and spectral images, including electron density (ED), iodine concentrations (IC), and effective atomic numbers (Zeff) and so on. In addition, SDCT can accurately measure the extracellular volume (ECV) by directly assessing the iodine density value [12,15,16].

Dual-energy CT has demonstrated promising results in evaluating tumor characteristics and treatment effectiveness [17,18,19,20,21]. IC, Zeff, and ED are surrogate measures to determine tumor vascularity and perfusion. IC directly quantifies iodine content, Zeff represents the average atomic numbers of the tissue, and ED reflects the probability of an electron occurring at a specific location influenced by the structure of the tissue molecule. These characteristics indirectly provide information about the accumulation of contrast agents, as in Ref. [22], particularly in diagnosing tumor lymph node metastasis, such as lung cancer, rectal cancer, head and neck cancer, etc. [7,22,23,24,25,26]. ECV is a composite of the extravascular, extracellular volume fraction, and intravascular space fraction, which is capable of detecting changes in the composition of the extracellular matrix [27]. ECV derived from SDCT is a practical and efficient method that eliminates the need for pre- and post-contrast image registration, which facilitates the application of ECV in various fields [28]. The tumor cells disrupt the natural structure of lymph nodes and expand the spaces blood vessels and surrounding tissues [29,30]. This expansion leads to contrast agent aggregation, which has been found capable of detecting metastasis of cervical lymph nodes in papillary thyroid carcinoma [31]. Therefore, using SDCT for lymphoma assessment both before and after treatment could be considered an economical, less exposed to radiation, easier to obtain, and efficient method if it could be proven to be a valid tool for determining the activity of lymph nodes in lymphoma.

Based on the above, this study was designed to explore the potential value of SDCT-derived parameters, such as IC, Zeff, ED, and ECV, in evaluating lymph node activity in lymphoma using ^18^FDG-PET/CT and the Deauville five-point scale (5-PS) as diagnostic criteria. We hypothesized a strong correlation between angiogenesis and metabolism in lymph nodes, which could be captured by SDCT and ^18^FDG-PET/CT. We aimed to develop alternative imaging biomarkers for measuring lymph node activity in lymphoma.

## 2. Materials and Methods

### 2.1. Study Design

The enrollment period was from April 2021 to April 2022. The inclusion criteria were: (1) adult patients pathologically confirmed with malignant lymphoma; (2) contrast-enhanced SDCT and ^18^FDG-PET/CT scans of head, neck, chest, abdomen, and pelvis, performed at an interval of less than one month, without interim lymphoma treatment; (3) patients without any other neoplasms or cancers. Fifty-five patients were included, and relevant clinical data were obtained retrospectively from medical records. The process for including and excluding patients is illustrated in Figure 1.

On the venous phase of contrast enhancement SDCT, we evaluated the size and spectral characteristics of lymph nodes in patients with pathologically diagnosed lymphoma, including IC, nIC, ED, nED, Zeff, nZeff, ECV, long-axis diameter, and short-axis diameter. According to the ^18^FDG-PET/CT and 5-PS criteria, a Deauville score > 3 was defined as an active lymph node [3].

### 2.2. 18FDG-PET/CT

All ^18^FDG-PET/CT images were acquired using a PET/CT scanner (Discovery VCT, GE Healthcare, Milwaukee, WI, USA) at the Department of Nuclear Medicine, Union Hospital, Tongji Medical College, Huazhong University of Science and Technology. Subjects were instructed to fast for at least six hours, ensuring their blood glucose levels were below 8 mmol/L before receiving an intravenous injection of 5 MBq/kg of ^18^F-FDG. The scanning process began 60 min after the injection and covered the area from the base of the skull to the upper thigh. Low-dose CT data were used to correct image attenuation, and a three-dimensional iterative reconstruction algorithm was used to merge the data with CT images.

### 2.3. SDCT

The patients underwent a contrast-enhanced CT examination using a 64-section SDCT system (IQon Spectral CT, Philips Healthcare, Amsterdam, The Netherlands). The scan covered the patient’s head, neck, chest, abdomen, and pelvis. The detailed acquisition parameters were as follows: tube voltage, 120 kVp; helical pitch, 0.8; rotation time, 0.5 s; and detector collimation at 64 × 0.625 mm^2^. The patients received an injection of a non-ionic iodinated contrast agent (350 mg/mL, iohexol, Accupaque 350, GE Healthcare, Boston, MA, USA) with a dose of 1.35 mL/kg at a rate of 3 mL/s followed by a 30 mL saline flush using a powered syringe (OptiVantage, Medtronic Covidien, Shanghai, China). The portal venous phase was acquired 60 s after contrast agent injection completion. The conventional images were reconstructed using the iDose algorithm, and the spectral images were reconstructed using the spectral reconstruction algorithm from the spectral-based imaging (SBI) data in a vendor-provided workstation (IntelliSpace Portal v9, Philips Healthcare) with a thickness of 1 mm, a section increment of 1 mm.

### 2.4. 18FDG-PET/CT Image Interpretation

We meticulously labeled and organized all assessed lymph nodes from each individual based on visual assessment, to ensure anatomic correlation and consistent measurements of lymph nodes between ^18^FDG-PET/CT and SDCT. Two radiologists, one with a decade of expertise (T.Y.) and the other with 15 years of experience (J.G.), carefully labeled all the lymph nodes and recorded their measurements independently. During the evaluation process, the lymph nodes were checked from the patient’s head to their pelvis, excluding any metastases outside the nodes in the liver, spleen, and bone marrow. The precise location of the lymph nodes was recorded using the 3D Slicer open-source software version 5.4.0 (https://www.slicer.org) [32].

The maximum standardized uptake value (SUV_max_) was used to measure the activity of lymph nodes. SUV_max_ of the liver and the mediastinal blood pool was determined. Regarding the Cheson criteria, a maximum of six lymphoma lesions per patient was assessed, with the largest and most well-defined lymph nodes chosen for measurement [33]. We also evaluated all the measurable lymph nodes that showed no activity on ^18^FDG-PET/CT. The evaluation of lymph node activity was based on the 5-PS criteria [34]: score 1, no FDG uptake above background; score 2, FDG uptake ≤ mediastinum uptake; score 3, FDG uptake > mediastinum uptake but ≤liver uptake; score 4, FDG uptake moderately > liver uptake; and score 5, FDG uptake markedly higher than liver uptake and/or new lesions were present. Lymph nodes that score less than or equal to three points indicate that lymph node activity has been inhibited, reflecting effective treatment. On the other hand, scores of four or five points indicate that the lymph node still has metabolic activity. The assessed lymph nodes were categorized into two groups: the active group and the inactive group. The active group comprised lymph nodes that exhibited anomalous FDG uptake before treatment and those with Deauville scores of four or five following treatment. Conversely, the inactive group included normal lymph nodes before treatment and those that decreased to a Deauville score of three or lower after treatment.

### 2.5. SDCT Quantitative Analysis

IC, nIC, Zeff, nZeff, ED, nED, and ECV values of the assessed lymph nodes were recorded on the workstation (IntelliSpace Portal Version 11, Philips Healthcare, Amsterdam, The Netherlands). No image registration was necessary since all spectral images were reconstructed from the original SBI data. During the study, two experienced radiologists (T.Y., with 10 years of experience, and J.G., with 15 years of experience) used the 3D Slicer software version 5.4.0 (https://www.slicer.org) to measure the target lymph nodes. We chose the largest central slice of the lymph node for delineation. An oval-shaped region of interest (ROI) was drawn manually to cover the entire lymph node while avoiding blood vessels, necrosis, calcification, and cystic degeneration by the other two radiologists (H.C., with 3 years of experience, and Y.F., with 7 years of experience) (Figure 2). By copying and pasting the ROI from one spectral image to another, we could avoid bias from inconsistent manual delineation and obtain accurate measurements. We obtained the mean values for IC, ED, and Zeff and the long-axis and short-axis diameters of lymph node ROIs. Additional ROIs were placed on the aorta or subclavian artery in the corresponding image slice as the target lymph node, in order to calculate relative values of IC, Zeff, and ED normalized to the contrast enhancement. The formulas were established as follows:(1)$$nIC={IC}_{lymphnode}/{IC}_{aorta},$$

IC_aorta_ refers to the IC of the aorta in the same image slice as the lymph node, the IC_lymph node_ is the IC of the target lymph node, and the normalized IC of the lymph node is represented by nIC;
(2)$$nZeff={Zeff}_{lymphnode}/{Zeff}_{aorta},$$

Zeff_aorta_ refers to the Zeff of the aorta in the same image slice as the lymph node, Zeff_lymph node_ is the Zeff of the target lymph node, and the normalized Zeff of the lymph node is represented by nZeff;
(3)$$nED={ED}_{lymphnode}/{ED}_{aorta},$$

ED_aorta_ refers to the ED of the aorta in the same image slice as the lymph node, the ED_lymph node_ is the ED of the target lymph node, and the normalized ED of the lymph node is represented by nED;
(4)$$ECV={[IC}_{lymphnode}-100\times hemotocrit(\%)]/{IC}_{aorta}$$

**Figure 2 diagnostics-14-00149-f002:**
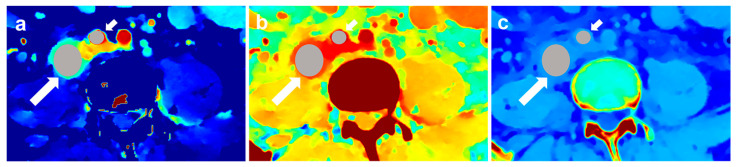
Exemplary illustration of ROI placement. Iodine map (**a**), effective atomic number map (**b**), and electron density map (**c**) are presented. The brighter the color, the higher the value it represents. The long arrow points to the target lymph node, and the short arrow points to the artery on the same slice.

### 2.6. Statistical Analysis

To compare continuous variables in the data set, we used the *t*-test for normally distributed variables and the Mann–Whitney U test for non-normally distributed variables. We determined the inter-observer agreement for quantitative variables using the intraclass correlation coefficient (ICC). Since each patient can have multiple lymph nodes, we used the generalized linear mixed model (GLMM) based univariate binary logistic regression model to identify statistically significant predictive variables and obtained each variable’s prediction probability and odd ratio (OR) value. We used the area under the receiver operating characteristic curve (AUROC) value to calculate the diagnostic efficacy of the spectral parameters. All statistical analyses were performed using the SPSS software (version 26.0, IBM Corp., Armonk, NY, USA) and R software (version 4.2.1, Boston, MA, USA). We considered two-tailed *p*-values < 0.05 to be statistically significant.

## 3. Results

### 3.1. Patient Characteristics

A total of 55 patients were included in this study, comprising 30 males and 25 females with a mean age of 52 ± 14 years. The pathological results of the included cases were as follows: 1 case of classical Hodgkin’s lymphoma, 36 cases of DLBCL, 5 cases of follicular lymphoma, 5 cases of mantle cell lymphoma, 2 cases of NK-T cell lymphoma, and 1 case of other types of lymphoma. Twenty-one patients had prior chemotherapy and 34 were newly diagnosed. Of the 289 lymph nodes, 236 were active and 53 were inactive. Baseline characteristics are listed in Table 1.

### 3.2. Between-Group Comparison and Diagnostic Performance of Spectral CT Parameters 

The spectral parameters and size of active and inactive lymph nodes are presented in Table 2. The nIC, nZeff, nED, ECV, and the long-axis and short-axis diameters of lymph nodes in the active group were significantly higher than those in the inactive group (*p* < 0.001) (Table 2). GLMM analysis was performed separately for nIC, nED, nZeff, ECV, and the long-axis and short-axis diameters of lymph nodes. The estimated values of the fixed-effect parameters for nIC, nZeff, and ECV were statistically significant (*p* < 0.05) (Table 3). Univariate binary logistic regression analysis of nIC, nZeff, ECV, and the long-axis and short-axis diameter of lymph nodes showed that the OR values were 21.93 (95%CI: 10.31–46.64), 30.35 (95%CI: 12.82–71.90), 20.17 (95%CI: 8.63–47.11), 13.61 (95%CI: 6.10–30.38), and 16.88 (95%CI: 7.74–36.81), respectively (Table 4). AUROC values of nIC, nZeff, ECV, the long-axis and short-axis diameters of lymph nodes were 0.822 (95%CI: 0.785–0.905), 0.845 (95%CI: 0.785–0.905), 0.811 (95% CI: 0.749–0.873), 0.778 (95%CI: 0.712–0.845), and 0.803 (95%CI: 0.737–0.869), respectively. Figure 3 shows one example of an inactive lymph node and three examples of an active lymph node.

### 3.3. Inter-Observer Agreement

The two radiologists’ measurements were consistent and showed a high level of agreement, including IC, nIC, ED, nED, Zeff, and nZeff, and the long-axis and short-axis diameters of lymph nodes, as well as IC, ED, and Zeff of the aorta or subclavian artery in the same image slice as the lymph node. The ICC values for all measurements were above 0.889 (Table 5).

## 4. Discussion

In this study, we revealed the potential clinical value of quantitative metrics from SDCT in evaluating the activity of lymph nodes of patients with lymphoma using ^18^ FDG-PET/CT as a reference. Both nIC and nZeff values of lymph nodes in the active group are higher than those in the inactive group. We found that nIC, nZeff, and ECV have a noteworthy impact on the assessment of lymph node activity. This discovery offers valuable insights that can greatly influence clinical practice.

Most lymphomas (93–99%) are FDG avid, making ^18^FDG-PET/CT the most important image modality in the assessment of patients with lymphoma. In contrast to conventional CT, which primarily relies on assessing abnormal lymph nodes based on size, PET-CT can offer valuable functional and metabolic insights, such as lymph node activity. The glycolytic demand of tumors and the high perfusion associated with tumor angiogenesis jointly establish the biological foundation of ^18^FDG-PET/CT [35,36] and recent research has established a significant correlation between metabolic activity, as assessed through ^18^FDG-PET/CT, and IC, as assessed via SDCT [37], suggesting the potential complementary role of SDCT in integrating information on metabolism, perfusion, and anatomy.

Our study reinforces the conclusion that nIC, nZeff, and ECV could effectively identify lymph node activity, aligning with prior findings [7,18,22,23,24,26,31,38]. An increase in the number of blood vessels and alterations in tumor-associated vascular patterns within malignant lymph nodes may contribute to the elevation of IC [39,40], resulting in higher Zeff values [38]. Although ^18^FDG-PET/CT is central to lymphoma assessment, high-resolution contrast-enhanced CT may be required for precise measurements of lymph nodes or mass size. As a result, SDCT could serve as a more effective complement to ^18^FDG-PET/CT by integrating information on metabolism, perfusion, and anatomy [41].

While previous research emphasized the role of the short axis in distinguishing between benign and malignant lymph nodes [24,25], our study found that it was not the most significant factor. We speculate that the different findings about the lymph node’s short axis in our study may be due to the inclusion of patients before and after treatment. We observed that after treatment, the activity of the lymph nodes was inhibited, but their morphology was not reduced. This suggests that functional changes in the lymph nodes may occur before any morphological changes following lymph node treatment. It is common for larger tumors to shrink, but smaller drug-resistant tumor clones may still be present even after the tumor has shrunk [42]. Lymph node masses can respond well to chemotherapy but may remain large due to fibrosis [10].

Based on our research, we discovered that the SUV_max_ values of various lymph nodes affected by lymphoma in a single patient were very similar. This suggests a potential “cluster effect” that could result in biased findings. Additionally, the number of lymph nodes affected by cancer varied from patient to patient. However, previous studies did not consider the correlation of multiple lesions in the same body [15,16,21,24,29,39,43]. While individual studies have mentioned these factors, their statistical analysis did not account for their impact [30]. After considering the two factors mentioned above, we utilized the GLMM model to correct the number of lymph nodes and eliminate interactions between different lymph nodes in the same individual. This helped to solve any possible influence caused by the different number of measured lymph nodes in the same patient [39]. After accounting for the “cluster effect,” we achieved relatively high diagnostic performance in evaluating lymph node activity (the AUROC for nIC was 0.822, for nZeff was 0.845, and for ECV was 0.811). This suggests the potential utility of SDCT in clinical lymphoma assessment.

SDCT is an emerging technique that employs two detector layers to simultaneously capture low- and high-energy data in all patients, using standard CT protocols to derive various quantitative metrics, including IC, Zeff, and ED. This allows for the assessment of incidentally discovered findings and offers several advantages: (1) improved contrast visualization; (2) artifact reduction; (3) material decomposition capability; and (4) reduced radiation dose [44]. The image quality improvement and high sensitivity of CT led to good agreement between the two observers for the measurements in our study. In addition, these spectral parameters are readily available on most commercial software platforms other than traditional HU. While previous studies have used a variety of imaging features, such as the slope of the spectral HU curve, VMI, or radiomics features extracted from parametric maps to explore the diagnostic efficacy, the advantage of our approach lies in its simplicity and easy accessibility, using only off-the-shelf commercial parameters without additional data postprocessing. The findings of this study, combined with the convenient availability of spectral post-processing data, present compelling evidence using nIC, nZeff, and ECV to evaluate lymph node activity in lymphoma patients can provide valuable insights, particularly in the area of personalized medicine. First, it can cost-effectively benefit the clinical decision. Investigating the integration of nIC, nZeff, and ECV into the current diagnostic system can facilitate lymphoma staging and treatment response with lower costs. In particular, SDCT would be a better way to identify treatment response or resistance in longitudinal follow-up than PET/CT during and after treatment. Thus, to explore the potential for a multimodal approach involving SDCT, parameters and other molecular profiling data might enhance the accuracy of predicting treatment response and improve patient stratification and treatment selection. This methodology holds great promise in the realm of personalized medical treatment and offers substantial clinical benefits.

However, certain areas in this study could be improved. First, this study included various types of lymphoma, creating heterogeneity in the cohort. However, this heterogeneity increases the generalizability of our results among different lymphoma subtypes somewhat. Second, our study’s relatively small number of active lymph nodes prevented subgroup analyses of cases before and after lymphoma treatment. Third, the relatively small number of active lymph nodes in our study limited subgroup analyses, and the inclusion of various types of lymphoma with imbalanced numbers requires further investigation with larger sample sizes. Fourth, all lymph nodes in our study were measured manually, and no automated or semiautomated measures of artificial intelligence or deep learning were used, which might induce measurement bias and a heavy workload. Finally, this study did not provide the threshold of nIC, nZeff, and ECV in diagnosing active lymph nodes in lymphoma, which needs further exploration by external validation of large samples.

## 5. Conclusions

In conclusion, using nIC, nZeff, and ECV as indicators of lymph node activity in patients with lymphoma might be a reliable and effective tool for diagnosis and treatment. With continued research and development, we hope to see even greater advancements in diagnosing and treating malignant lymphoma through SDCT, especially as a complementary of ^18^FDG-PET/CT.

## Figures and Tables

**Figure 1 diagnostics-14-00149-f001:**
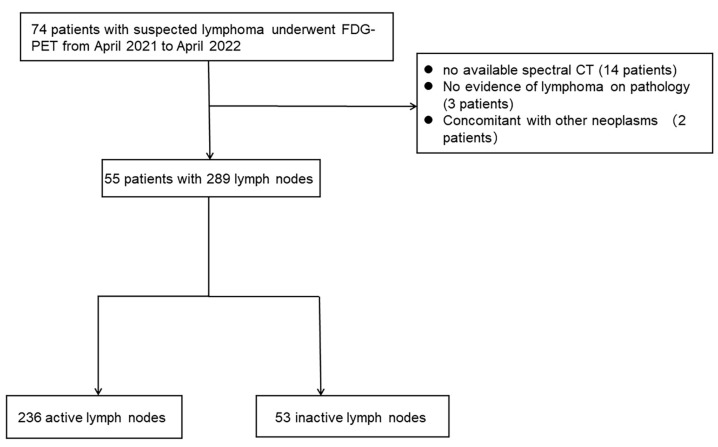
Flowchart shows the study population.

**Figure 3 diagnostics-14-00149-f003:**
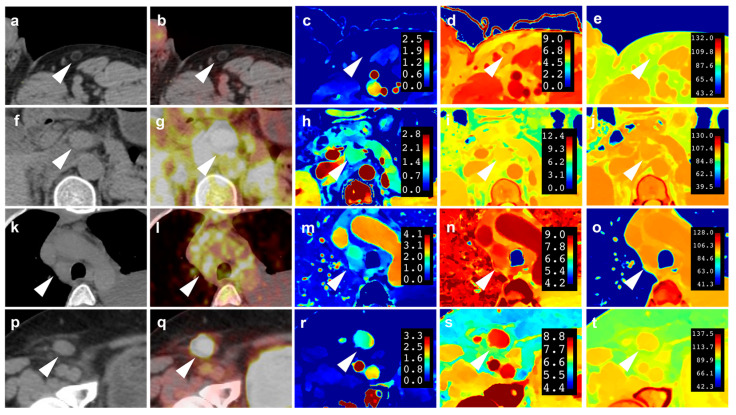
Images of inactive lymph nodes (case 1, (**a**–**e**)) and active lymph nodes (case 2, DLBCL, (**f**–**j**); case 3, marginal zone lymphoma, (**k**–**o**); case 4, follicular lymphoma, (**p**–**t**)). Axial non-contrast CT (**a**,**f**,**k**,**p**) and fusion images of PET and CT (**b**,**g**,**l**,**q**) are presented. The SUV_max_ of the inactive and active lymph nodes (white arrowheads) are 0.00 g/mL (**b**), 9.80 g/mL (**g**), 2.2 g/mL (**l**), and 15.4 g/mL (**q**). Iodine maps in the venous phase are presented (**c**,**h**,**m**,**r**). The iodine concentrations of the inactive and active lymph nodes (white arrowheads) are 0.36 mg/mL (**c**), 2.27 mg/mL (**h**), 1.37 mg/mL (**m**), and 1.43 mg/mL (**r**). The effective atomic number maps in the venous phase are presented (**d**,**i**,**n**,**s**). The effective atomic number of the inactive and active lymph nodes (white arrowheads) are 7.51 (**d**), 8.50 (**i**), 8.09 (**n**), and 8.12 (**s**). The electron density maps in the venous phase are presented (**e**,**j**,**o**,**t**). The electron density values of the inactive and active lymph nodes (white arrowheads) are 98.2 × 10^23^/cm^3^ (**e**), 103.8 × 10^23^/cm^3^ (**j**), 104.3 × 10^23^/cm^3^ (**o**), and 103.8 × 10^23^/cm^3^ (**t**).

**Table 1 diagnostics-14-00149-t001:** Basic characteristics of patients and lymph nodes.

Characteristics	Title 2
Patient age (y)	
Mean ± SD (range)	52 ± 14 (18–83)
Gender	
Male	30 (55)
Female	25 (45)
FDG PET/CT results	
active lymph nodes	236 (82)
inactive lymph nodes	53 (18)
Pathological results	
Hodgkin’s lymphoma	1 (2)
Non-Hodgkin’s lymphoma	
Diffuse large B cell lymphoma	36 (65)
Follicular lymphoma	5 (9)
Mantle cell lymphoma	5 (9)
Peripheral T Cell Lymphoma	1 (2)
Angioimmunoblastic T-cell lymphoma	1 (2)
Marginal zone lymphoma	1 (2)
NK-T cell lymphoma	2 (4)
MALT lymphoma	1 (2)
High grade B cell lymphoma	1 (2)
Anaplastic large cell lymphoma	1 (2)

Notes: Quantitative data were presented as mean ± standard deviation (range); categorical data were presented as count with percentage. Some percentages do not total 100 because of rounding.

**Table 2 diagnostics-14-00149-t002:** Comparison of dual-layer spectral CT parameters of active lymph nodes and inactive lymph nodes.

Parameters	Active Lymph Nodes (42 Patients, 236 Lymph Nodes)	Inactive Lymph Nodes (26 Patients, 53 Lymph Nodes)	*p* Value
IC (mg/mL)	1.29 ± 0.51	0.45 ± 0.39	<0.001 ^b^
Zeff	8.03 ± 0.27	7.28 ± 0.46	<0.001 ^b^
ED (×10^23^/cm^3^)	103.50 ± 6.49	99.21 ± 3.31	<0.001 ^b^
nIC	0.34 ± 0.14	0.11 ± 0.10	<0.001 ^b^
nZeff	0.88 ± 0.04	0.79 ± 0.05	<0.001 ^a^
nED	0.99 ± 0.06	0.95 ± 0.03	<0.001 ^b^
ECV	20.59 ± 12.41	5.58 ± 6.74	<0.001 ^b^
Long-axis diameter (mm)	19.37 ± 4.58	13.65 ± 3.38	<0.001 ^b^
Short-axis diameter (mm)	14.24 ± 4.10	8.69 ± 2.84	<0.001 ^b^

Notes: Data are presented as mean ± standard deviation. Abbreviations: IC, iodine concentration; Zeff, effective atomic number; ED, electron density; nIC, normalized iodine concentration; nZeff, normalized effective atomic number; nED, normalized electron density; ECV, extracellular volume. ^a^ Data were compared using a two-sample *t*-test. ^b^ Data were compared using the Mann–Whitney U test.

**Table 3 diagnostics-14-00149-t003:** Summary of generalized linear mixed model results for dual-layer spectral CT parameters.

Parameters	Estimated Coefficient	95%CI	*p* Value
nIC	91.28	26.68–155.88	0.006
nZeff	135.49	62.79–208.19	<0.001
nED	3.94	−2.25–10.13	0.21
ECV	0.72	0.13–1.31	0.02
Long-axis diameter (mm)	0.67	0.35–0.99	<0.001
Short-axis diameter (mm)	0.93	0.51–1.36	<0.001

Abbreviations: nIC, normalized iodine concentration; nZeff, normalized effective atomic number; nED, normalized electron density; ECV, extracellular volume.

**Table 4 diagnostics-14-00149-t004:** Summary of univariate binary logistic analysis for dual-layer spectral CT parameters.

Parameters	OR	95%CI	*p* Value
nIC	21.93	10.31–46.64	<0.001
nZeff	30.35	12.82–71.90	<0.001
ECV	20.17	8.63–47.11	<0.001
Long-axis diameter (mm)	13.61	6.10–30.38	<0.001
Short-axis diameter (mm)	16.88	7.74–36.81	<0.001

Abbreviations: OR, odds ratio; nIC, normalized iodine concentration; nZeff, normalized effective atomic number; ECV, extracellular volume.

**Table 5 diagnostics-14-00149-t005:** Inter-observer agreement analysis.

Parameters	Agreement (95%CI)
IC	0.889 (0.861–0.911)
Zeff	0.907 (0.884–0.925)
ED	0.940 (0.910–0.958)
IC_aorta_	0.961 (0.948–0.970)
Zeff_aorta_	0.935 (0.919–0.948)
ED_aorta_	0.998 (0.998–0.999)
nIC	0.913 (0.889–0.931)
nZeff	0.959 (0.949–0.968)
nED	0.942 (0.919–0.958)
ECV	0.928 (0.910–0.942)
Long-axis diameter (mm)	0.910 (0.888–0.928)
Short-axis diameter (mm)	0.912 (0.891–0.930)

Notes: Values are expressed as ICC for quantitative parameters; 95%CIs are in parentheses. Abbreviations: IC, iodine concentration; Zeff, effective atomic number; ED, electron density; IC_aorta_, iodine concentration of aorta; Zeff_aorta_, normalized effective atomic number of aorta; nIC, normalized iodine concentration; nZeff, normalized effective atomic number; nED, normalized electron density; ECV, extracellular volume.

## Data Availability

The anonymous data presented in this study are available on request from the corresponding author. The data are not publicly available due to institutional regulations.

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
