# Peer review of "Dual-Layer Spectral Detector Computed Tomography Quantitative Parameters: A Potential Tool for Lymph Node Activity Determination in Lymphoma Patients"

_diagnostics, 2024, doi:10.3390/diagnostics14020149_

Round 1

Reviewer 1 Report

Comments and Suggestions for Authors

1. Figure 2 caption, "(f, g) Iodine maps ......" should be "(c, g) Iodine maps". Suggest adjusting the contrast of these two panels to make the lymph nodes more visible (brighten up the images).

2. In the abstract, can you motivate the use of CE CT to detect lymph node activity in this group of patients as opposed to the reference-standard PET/CT, in a sentence or two. 

Reviewer 2 Report

Comments and Suggestions for Authors

Dear author,

I am writing to provide my review of your recent article on the use of contrast-enhanced dual-layer spectral detector CT (SDCT) in assessing lymph node activity in lymphoma patients. This letter reflects my evaluation of your study, focusing on its originality, methodology, and implications.

First and foremost, I commend you and your team for undertaking a study on such an original topic. The comparison of SDCT with 18FDG-PET/CT and the Deauville five-point scale in this context is highly innovative and addresses a critical aspect of lymphoma diagnosis.

Your retrospective analysis involving 55 patients and 289 lymph nodes offers significant insights, particularly the higher values of normalized iodine concentration (nIC), normalized effective atomic numbers (nZeff), normalized electron density (nED), and extracellular volume (ECV) in active lymph nodes. These findings are indeed intriguing and may have a considerable impact on clinical practices.

The application of generalized linear mixed models and receiver operating characteristic (ROC) analysis in your study adds a level of rigor to your findings. The AUROC values for nIC, nZeff, and ECV are promising, suggesting these could serve as reliable indicators of lymph node activity.

I would like to offer some suggestions to further enhance the value of your work:

  1. Clarification in Table 5: For Table 5, it would be beneficial to clearly specify when standard deviation and when a 95% range is being used. This clarity would help readers better understand and interpret the data.

  2. Moderation in Discussion: While the findings are interesting, I recommend toning down the discussion regarding the superiority of SDCT over traditional dimension measurements. Based on your results, it would be more appropriate to state that SDCT offers interesting insights but it is not conclusively proven to be superior to conventional measurements. Acknowledging this in the discussion would provide a more balanced view of the study's implications.

  3. Broader Context and Future Research: Exploring how these findings could influence treatment decisions, particularly in personalized medicine, would offer valuable insights. Additionally, suggesting potential future research directions, such as longitudinal studies or involving larger patient cohorts, would be beneficial.

In summary, your article is a significant contribution to lymphoma diagnostics, with SDCT showing great potential as a diagnostic tool. Your research establishes an excellent foundation for further studies and potential clinical application.

Thank you for the opportunity to review your insightful and important work.

Sincerely,

Reviewer 3 Report

Comments and Suggestions for Authors

Brief summary

The authors aimed to assess the value of different quantitative parameters obtained from a dual-layer spectral CT technology in the characterization of lymph nodes in lymphedema patients, compared with data obtained form PET CT to assess the lymph nodes activity.

TOPIC

Medium/high interest

TITLE

change as recommended in the attached file

INTRODUCTION

The description of different study is difficult to read and to follow and needs to be rewritten in a clear and fluid way

List only the quantitative analyisis related to the study in the introduction (including virtual non-contrast (VNC), virtual monoenergetic images (VMI), eq....)

MATERIALS

The case series is quite limited and inhomogeneous, as it includes diffent types of Lymphoma. This aspect should be highlighted in the limitation of the study

The procedure to obtain the different quantitative values should be explaining thorugh images showing the drawning of the ROIs

Discussion

Highlight the value of the study in the clinical pratice and in the patients'managment

Figures

More cases are needed to support the study results: moreover, the correlations with CT-PET images should be displayed to support the results

The authors should insert more images and more cases, and cases with differnt lymphoma histologies

Avoid the use of too many abbreviations

See Other changes in the attached file

There is interest in the application of DECT Technology, but the article should be partially rewritten to be suitable for publication

  • Is the manuscript clear, relevant for the field and presented in a well-structured manner? No, some section needs to be rewritten for clarity
  • Are the cited references mostly recent publications (within the last 5 years) and relevant? Does it include an excessive number of self-citations? Some new references are needed
  • Is the manuscript scientifically sound and is the experimental design appropriate to test the hypothesis?YEs
  • Are the manuscript’s results reproducible based on the details given in the methods section? More information about the ROI drawing are required. More cases are needed to support the results
  • Are the figures/tables/images/schemes appropriate? Do they properly show the data? Are they easy to interpret and understand? Is the data interpreted appropriately and consistently throughout the manuscript? Please include details regarding the statistical analysis or data acquired from specific databases. More cases are needed and comparison with PET CT images
  • Are the conclusions consistent with the evidence and arguments presented? An highligh of the limitations of the study is needed
  •  
  • Please evaluate the ethics statements and data availability statements to ensure they are adequate. ok

Comments on the Quality of English Language

Long pariods, especially in the introduction, needs to be rewritten for clarity

Try to provide more fluency in the introduction

Round 2

Reviewer 3 Report

Comments and Suggestions for Authors

I thank the authors for performing the suggested revisions